# Deformation of Gels with Spherical Auxetic Inclusions

**DOI:** 10.3390/gels8110698

**Published:** 2022-10-29

**Authors:** Jan Zidek, Petr Polacek, Josef Jancar

**Affiliations:** Central European Institute of Technology (CEITEC), Brno University of Technology, Purkynova 123, 612 00 Brno, Czech Republic

**Keywords:** auxetic, deformation, model, simulation, continuum mechanics

## Abstract

Auxetic metamaterials possess unnatural properties, such as a negative Poisson’s ratio, which offers interesting features when combined with traditional materials. This paper describes the deformation behavior of a gel consisting of spherical auxetic inclusions when embedded in a conventional matrix. The auxetic inclusions and conventional matrix were modeled as spherical objects with a controlled pore shape. The auxetic particle had a reentrant honeycomb, and the conventional phase contained honeycomb-shaped pores. The deformation behavior was simulated using various existing models based on continuum mechanics. For the continuum mechanics models—the simplest of which are the Mori–Tanaka theory and self-consistent field mechanics models—the auxetic particle was homogenized as a solid element with Young’s modulus and Poisson’s ratio and compared with the common composite gel filled with rigid spheres. The finite element analysis simulations using these models were performed for two cases: (1) a detailed model of one particle and its surroundings in which the structure included the design of both the reentrant and conventional honeycombs; and (2) a multiparticle face-centered cubic lattice where both the classic matrix and auxetic particle were homogenized. Our results suggest that auxetic inclusion-filled gels provide an unsurpassed balance of low density and enhanced stiffness.

## 1. Introduction

Porous hydrogels are materials that are frequently used in medicine [1,2,3]. The mechanical response of hydrogels is an important parameter of such materials; it is a part of the mechanical engine in which hydrogel deformation is tuned with other components, such as bones, muscles, and tissues. The mechanics can be tuned by the shape of the pores and their distribution in the sample. This paper presents a model wherein gels are tuned with auxetic cells as pores, which means a material with unusual deformation in a perpendicular sense to the specific direction of loading.

Several methods of improving the mechanical performance of gels are described in the literature. Mechanical performance refers to increased stiffness, i.e., Young’s modulus and shape stability during deformation. The basic method is additional covalent crosslinking [4,5], which increases the modulus of a macromolecular network. Mostly, additional crosslinking significantly changes the behavior of a hydrogel, which is unwanted. The next method of improving mechanical performance is mixing with a filler, which can be a powder of rigid particles. The disadvantage of a classic rigid filler composite is the high density of the composite gels. Recent production strategies of lightweight composites are primarily based on the design of nanocomposites. The reinforcing effect of nanocomposites is achieved with a smaller fraction of filler than in classical composites [6,7,8,9]. The next strategy is designing syntactic foams in which the solid filler is substituted with hollow particles [10,11,12,13]. The next possible strategy is proposed in this paper: a composite of a classic matrix and auxetic inclusions. We found that auxetic inclusions provide a reinforcing effect while simultaneously being lightweight. Such a composite may be an alternative to classic particle composites in which the positive effect of reinforcement is reached at the expense of increased density.

This paper aims to provide a detailed description of the micromechanics of these composites. The models from micromechanics describe mechanical behavior in terms of continuum mechanics. The micromechanics of the model comprises materials with conventional and auxetic inclusions are studied. The conventional porous material is homogenized as a continuous element with a given modulus and positive Poisson’s ratio. The auxetic array of pores can be homogenized as an element with a negative Poisson’s ratio. The models from micromechanics elucidate the underlying reasons for the auxetic inclusion having a reinforcing effect. Numerical simulations enable us to compare the deformation behavior of standard composites with rigid particles and gels that include auxetic particles. The word auxetic—meaning “incremental” in Greek—denotes materials or structures that have a negative Poisson’s ratio. Materials with a negative ratio of lateral contraction have been reported in early publications on particle mechanics [14] and have been systematically investigated since the 1990s [15,16,17,18]. The auxetic materials belong to the group of metamaterials with optical, electromagnetic functions (Appendix A).The first types of auxetic materials were structured in 2D auxetic mats, which were auxetic in one direction of deformation. Research on auxetic materials was predominantly focused on theoretical models [19,20]. The next phase was 3D modeling, in which one direction was stretched or compressed and both of the lateral directions were auxetic [21,22]. There has been a rapid rise in interest in auxetic materials since approximately 2015 when 3D printing became a widely available technology. The expansion of the use of printed auxetic materials then followed, and deformation was frequently analyzed by video recording [23,24,25,26]. Auxetic structures have rapidly evolved over the past three decades. However, discussions remain surrounding its appropriate applications, which were relatively rare in the textile industry [27], building industry [28], or in mechanical engineering industries in the form of dowels [29]. In recent years, additional applications in the field of medicine have also been proposed.

Specifically, attempts have been made to incorporate auxetic materials in tissue engineering [30,31]. However, the number of applications in which auxetic materials can be used remains small despite the intensive development in recent years. To date, the research on auxetic materials has been largely limited to descriptions of homogenous auxetic materials and geometric evolution during deformation [32,33,34]. Recently, the behavior of the traditional/auxetic material interface has attracted considerable interest [35], and this interface is important for several reasons. Parts of auxetic materials can be used as components of complex products and specialized units [20]. Auxetic parts must be mechanically integrated into the machine design, whereas the other components are conventional. Alternatively, auxetic materials are used as wound healing scaffolds [26,31,36], where the materials are attached to the surrounding tissue with a positive Poisson’s ratio [37]. Wound healing applications of auxetic materials require the inclusion of a healing gel, which is an incompressible viscous liquid (i.e., its Poisson’s ratio approaches 0.5). Therefore, furthering our knowledge of conventional/auxetic materials is important. These hybrid materials are supposed to have a specific function. For example, sandwich structures and conventional–auxetic–conventional layers are presented in the literature as ballistic materials [20]. The deformation of such materials produces increased tension at the interphase, which may be advantageous in certain applications because of the interphase absorbing a significant amount of mechanical energy. Some hybrid auxetic/conventional systems can provide interesting functionalities; for example, the KinetiX system [38] enables individuals to produce programmable materials and switch the auxetic function to selected holes in a targeted manner. The next application of the controlled mechanical response of hybrid auxetic materials is strain sensing, which is the collection of mechanical energy and transformation to electric energy; it is currently performed with gels that have filler, such as carbon black [39] or zinc oxide [40]. The auxetic enables the designer to achieve controllable high sensitivity [41,42].

This paper deals with the use of solid phase models from classical mechanics. Solid phase models provide a suitable justification for the reinforcing effect of auxetic particles. These models allow us to calculate the strain intensity tensor caused by certain types of filler inclusions. The oldest models of composites are the Voigt and Reuss rules. The Voigt and Reuss model is described by the Hashin–Shtrikman bounds [43]; it is the simplest model in the upper and lower boundary of the effective modulus of a composite. The models are not sufficient for an investigation of the effect of Poisson’s ratio because the bounds are independent of this particular ratio. In contrast, models based on micromechanics describe composite structures in terms of general elasticity, including Poisson’s ratio [44]. The analysis of the micromechanics of auxetic materials has been neglected in the literature to date. Current models based on micromechanics have been intensively studied on conventional particle composites and are frequently reported in the literature [44,45]. The basic principle of micromechanics is derived from Eshelby’s inclusion [46,47]. Eshelby studied the possible stress, strain, and displacement fields in elastic bodies containing elastic inclusions. The inclusion was subjected to transformation, which was limited by the surrounding material. In such instances, the inclusion and surrounding material remained stressed. This theory calculates the strain states of the body. Eshelby found that the strain and stress field inside a spherical inclusion is uniform and has a closed-form solution regardless of the material’s properties and initial transformation strain (also called the eigenstrain). Other approaches have also been based on Eshelby’s approach. The self-consistent mechanics (SCM) approach calculates the strain concentration tensor [48,49], which expresses the inclusion’s strain in terms of the dilute system’s externally applied strain. A homogeneous medium with the properties of the composite was assumed to surround the inclusion. The Mori–Tanaka model was used to calculate the average internal stress in a matrix of a material containing inclusions using eigenstrains from Eshelby’s theory [50,51]. This represented the starting point for the determination of the effective behavior of composites. The SCM and Mori–Tanaka (MT) methods are presented in detail below. In addition to the analytical methods of composite materials, there are many numerical models for conventional solid particle composite materials. The most widely used methods are based on the finite element method (FEM). A small computer limited the initial analyses, and the structure of the cell was significantly simplified into single-cell and multi-cell models. The quarter-cell approach modeled a smooth particle, and the structure of the composite was simplified to one quadrant. The FEM was applied to analyze the distribution of strain and stress in the composite material [52].

## 2. Results

Two material models are proposed in this article. An overview of the methods employed in this article is provided to further indicate the orientation of the research. All analyses describe a comparison of standard composites with rigid particles and with auxetic particles under otherwise equal conditions.

Three materials were compared (*E*: Young’s modulus; υ: Poisson’s ratio):Matrix: fully conventional sample (*E*_m_, υ_m_);Composite gel: conventional matrix filled with rigid particles (*E*_m_ ≪ *E*_p_, υ_m_ = υ_p_);Auxetic gel: conventional matrix filled with auxetic particles (*E*_m_
*= E*_p_, υ_m_ > 0, υ_p_ < 0).

Two analyses were performed:
Models based on the FEM (R, S);Models based on micromechanics (S).

Two structural approaches were applied (S and R):
(S)Solid phase modeling: The particles and matrix were modelled as a continuum, and the auxetic function was introduced by the prescription of a negative Poisson’s ratio to the solid object.(R)Real material design: A negative Poisson’s ratio was introduced by designing auxetic cells in the material, and the rigid particle was based on the reinforcement of the cells in the material.

### 2.1. Models from Micromechanics

#### 2.1.1. Mori–Tanaka Model

Let us define the material properties, polymer matrix modulus (*E*_m_), Poisson’s ratio of the matrix (υ_m_), particle modulus (*E*_i_), and particle Poisson’s ratio (υ_i_). The particle’s geometry was simply spherical, and the aspect ratio of the particle was equal to one.

The effective modulus of the material with the undeformable inclusion and auxetic inclusion was calculated according to the MT theory. The isotropic stiffness tensor for the inclusion (**s**_i_) and matrix (**C**_m_) was calculated from the generalized Hooke’s law: (1)$$C^{-1}=\left[\begin{array}{cccccc} \frac{1}{E} & -\frac{\upsilon }{E} & -\frac{\upsilon }{E} & 0 & 0 & 0\\ -\frac{\upsilon }{E} & \frac{1}{E} & -\frac{\upsilon }{E} & 0 & 0 & 0\\ -\frac{\upsilon }{E} & -\frac{\upsilon }{E} & \frac{1}{E} & 0 & 0 & 0\\ 0 & 0 & 0 & \frac{1}{G} & 0 & 0\\ 0 & 0 & 0 & 0 & \frac{1}{G} & 0\\ 0 & 0 & 0 & 0 & 0 & \frac{1}{G} \end{array}\right]$$
where E is the elastic modulus, υ is the Poisson’s ratio, and G is the shear modulus G = 0.5E/(1 + υ).

The compliance (**s**) was calculated as the inverse matrix of C:(2)$$s=C^{-1}$$

The Eshelby fourth-order tensor was calculated, which related the constrained strain inside the inclusion of its eigenstrain:
(3)$$S^{*}=\left[\begin{array}{cccccc} S_{1111}^{*} & S_{1122}^{*} & S_{1133}^{*} & S_{1112}^{*} & S_{1123}^{*} & S_{1131}^{*}\\ S_{2211}^{*} & S_{2222}^{*} & S_{2233}^{*} & S_{2212}^{*} & S_{2223}^{*} & S_{2231}^{*}\\ S_{3311}^{*} & S_{3322}^{*} & S_{3333}^{*} & S_{3312}^{*} & S_{3323}^{*} & S_{3331}^{*}\\ S_{1211}^{*} & S_{1222}^{*} & S_{1233}^{*} & S_{1212}^{*} & S_{1223}^{*} & S_{1231}^{*}\\ S_{2311}^{*} & S_{2222}^{*} & S_{2333}^{*} & S_{2312}^{*} & S_{2323}^{*} & S_{2331}^{*}\\ S_{3111}^{*} & S_{3122}^{*} & S_{3133}^{*} & S_{3112}^{*} & S_{3123}^{*} & S_{3131}^{*} \end{array}\right]$$


For the specific case of spherical inclusion, the elements of the Eshelby tensor were
(4)$$S_{1111}^{*}=S_{2222}^{*}=S_{3333}^{*}=\frac{7-5\upsilon }{15\left(1-\;\upsilon \right)}$$
(5)$$S_{1122}^{*}=S_{2233}^{*}=S_{3311}^{*}=S_{1133}^{*}=S_{2211}^{*}=S_{3322}^{*}=\frac{5\upsilon \;-1}{15\left(1-\;\upsilon \right)}$$
(6)$$S_{1212}^{*}=S_{2323}^{*}=S_{3131}^{*}=\frac{4-5\upsilon }{15\left(1-\;\upsilon \right)}$$

The next elements of the matrix were equal to zero.

The strain concentration tensor (**T**) was calculated as
(7)$$T={\left[I+S_{m}^{*}\;s_{m}\left(C_{i}-C_{m}\right)\right]}^{-1}$$
where **I** identifies a diagonal identity tensor of size 6 × 6.

The strain concentration tensor (Tc) of the composites with the volume fraction (φ) of the particle is calculated:(8)$$T_{C}=I\left(1-\varphi \right)+T\varphi$$

The elasticity polarization tensor P was calculated from the strain concentration tensor:(9)$$P_{C}=\varphi \left(C_{i}-C_{m}\right)T_{C}$$

The effective stiffness tensor of the composite was calculated by the formula:(10)$$C_{\mathrm{eff}}=C_{m}+P_{C}T_{C}^{-1}$$

#### 2.1.2. The Self-Consistent Mechanics Theory

In the initial step, let us say that the effective stiffness tensor is equal to the matrix tensor:(11)$$C_{eff,0}=C_{m}$$

The compliance tensors for the matrix and inclusion S_m_ and S_i_ were calculated by Equation (2), and the Eshelby matrices S_m_ and S_i_ were calculated by Equation (3).

The partial strain concentration tensors for the inclusion and matrix were calculated as follows:(12)$$T_{i}={\left[I+S_{i}^{*}\;s_{i}\left(C_{i}-C_{eff}\right)\right]}^{-1}$$
(13)$$T_{m}={\left[I+S_{m}^{*}\;s_{m}\left(C_{m}-C_{eff}\right)\right]}^{-1}$$

The composite strain concentration tensor was calculated as:(14)$$T=\varphi \;T_{i}+\left(1-\varphi \right)\;T_{m}$$

The elastic polarization tensor was, analogously,
(15)$$P=\varphi C_{i}\;T_{i}+\left(1-\varphi \right)C_{m}\;T_{m}$$

The effective stress after the first iteration was
(16)$$C_{\mathrm{eff},1}=P\;T^{-1}$$

The calculation was repeated from Equation (12) after the nth iteration:(17)$$\Delta C=C_{\mathrm{eff},n}-C_{\mathrm{eff},n-1}$$

The iteration cycles were calculated until the limiting condition was lower:(18)max(ΔC)<tol
where tol is an optional tolerance interval. For the models with a modulus of the matrix of approximately 2.5 kPa and a Poisson’s ratio of 0.39, the value of tol was selected as 10^−6^.

For more complex tasks such as anisotropic materials and more types of inclusions, the authors recommend the MMTensor package for MATLAB (programmed by Maarten Moesen, University, KU Leuven, Leuven, Belgium, EU) [53].

### 2.2. Simulations

This article describes a model of materials that contained spherical inclusions from an auxetic material in a conventional matrix. The models were analogously designed to the composites, with rigid, non-deformable particles in a soft matrix. The rigid particles were not deformed; however, their presence significantly affected the behavior of the matrix. An auxetic material is a material with a negative Poisson’s ratio. In other words, it is a material that, when stretched in one direction, simultaneously elongates in the perpendicular direction. When this material is compressed in the direction of one axis, it also becomes compressed in perpendicular directions.

Auxetic structures exist, and they can be created by the 3D printing technique. The model shown in Figure 1 is an example of an auxetic system with a simple structure; it is a spherical inclusion of a filler in the matrix. The advantage of the material lies in its material homogeneity; therefore, the load transfer from the matrix to the particle is not complicated by interphase effects, such as adhesion on the interphase or dissociation of the matrix and particle.

The advantage of such a honeycomb system is its superior control of macroscopic parameters, mainly Poisson’s ratio. Each shape of the honeycomb has its theoretical Poisson’s ratio. A regular hexagon honeycomb has a theoretical Poisson’s ratio of one [54]. The system can be switched from conventional to auxetic by selecting the local parameters of one cell. The 3D spatial structure of the honeycomb was created (Figure 1a) using this procedure, which is presented in the Appendix A. A 3D structure is more realistic than a 2D structure, as a reentrant 2D honeycomb plate combines the auxetic-normal response in perpendicular directions, whereas a 3D structure is auxetic in any direction of the xy plane. A slice from the 3D model is presented in Figure 1b. As mentioned above, the single-axis projection of a 3D model in either the x- or y-axis is a standard honeycomb plane. The reentrant cells are depicted in violet in the figure. The reentrant cells box has an auxetic response, while the box from regular honeycombs has a standard response.

As shown in Figure 1a,b, the model body was materially a continuous structure, and there was no transition between the material in the matrix and the particle. Only in the spherical inclusion was the material arranged in an auxetic structure and the matrix in a conventional honeycomb structure.

The structure shown in Figure 1 can be transformed into a conventional solid particle composite gel when setting a high modulus of elasticity for the inclusion material. The normal z-stress distribution in the specific case of the composite gel is shown in Figure 1c. This distribution is the stress acting in the direction of the z-axis on the plane with a z-normal vector.

The highest local stress was recorded in the cylindrical layer through the sample in the z-direction, limited by the xy cross-section of the particle. The load path between the upper and lower side of the sample passed directly through the center of the particle and was parallel to the direction of elongation. The load transfer through the particle was mostly observed in the particulate composites. In the case of a stiff particle with a high modulus, the material in Figure 1c was constituted as a composite gel, and its response was similar to that of a standard composite gel.

The model in Figure 1d was uniform in terms of materials. All of the elements had the same stiffness and Poisson’s ratio. The switch of the central domains to auxetic behavior was caused by pore geometry. The particle behaved in an auxetic way because it was formed from auxetic reentrant cells.

The local deformation inside the particle was significantly lower than the deformation of the matrix despite the fact that the stiffness of the matrix and inclusions were similar. When they formed a homogenous structure, the auxetic inclusion altered the behavior of the matrix in the composite gel material. The reason for this was the opposing forces acting at the interface between the particle and the matrix. The matrix tended to compress the particle in the perpendicular direction, but the particle tended to expand. These forces canceled each other out, resulting in a relatively low transverse deformation at this interface.

Figure 1d shows the z-stress distribution. The load was transferred from the upper wall subject through the particle to the lower wall by deformation. The majority of the load was transferred through the interphase layer, which meant that the most loaded elements were at the interphase between the particle and the matrix. However, the stress inside the particle was lower than the stress on the interface.

The section describes a model consisting of only one representative particle in the material. This model provided detailed information about the deformation and stress distribution in the matrix, particle, and interphase. The computing power of one PC workstation was sufficient to describe one particle and the surrounding matrix on the level of exact cells.

However, more exact data about the load transfer can be found in a model with multiple particles. In the case of multiple (172) particles, the model must be simplified. Thus, both the particle and the matrix were modeled as a continuum with a conventional or auxetic Poisson’s ratio.

The model presented in this article contained monodisperse particles regularly distributed in an FCC lattice. The FCC lattice was selected because its spatial distribution enables one of the highest sphere packing densities of monodisperse particles (an alternative is hexagonal sphere packing [HSP], which has the same packing density). The layer of a matrix between the two nearest particles is thicker in the cases of FCC and HSP than in any other case.

This lattice model can be more clearly interpreted than randomly distributed particles. The nearest particle distance (NPD) is the shortest distance between two neighboring particles. In an FCC lattice, the NPD can be unambiguously defined and calculated from analytical geometry. By contrast, the random particle model has a wide distribution of NPDs [55]. The models with random particle distributions contain close particle clusters. In these clusters, the stress is increased, and the load is transferred with priority by this short connection inside the cluster regardless of whether the particle is rigid or auxetic.

Most loads can be transmitted through NPD layers. An analysis of composite behavior must be based on the behavior of the matrix. A homogeneous matrix model with a Poisson’s ratio of zero was created as a reference model, and a Poisson’s ratio of zero was set for the matrix. The matrix material had a prescribed elongation. In this model, the load was uniform throughout the system such that all elements were loaded with the same stress. For 100% deformation, this stress was equal to 0.82 kPa, and it corresponded to the green color on the color scheme in the color map.

Next, the deformation of composite gels with 5%, 10%, 20%, and 30% volumes of particles was modeled. The results from the 30% model are presented in the main text, while the results of the remaining models can be found in the Appendix A. All models were deformed to the same total stress in the box (σ_z_ = 0.82 kPa). The matrix had a relative deformation of 30%. The same stress was observed at a stretching percentage of 14.5% for the auxetic material and 9% for the composite.

A conventional composite is a material comprising a matrix combined with nearly non-deformable spherical particles. The effects of non-deformable particles in a composite have been extensively described in the literature. Our results were comparable to the results published in the literature; namely, that the presence of particles affected the deformation of the matrix, and the highest local stress was observed in the layer of the matrix between the nearest particles. The highest z-normal stress was observed at the intersection between the nearest particles (Figure 2a). The distribution of z-normal stress for all volume fractions are presented in Appendix A. The Animation of evolution of z-stress during the stretching is in Appendix A.

The structure of the new combined material based on the composite differed in its behavior even though the initial spherical spatial configuration was identical to that of the composite. The matrix and particles were composed of elements of the same modulus. The different parameter between the particles and the matrix was the sign of Poisson’s ratio. A particle with a negative Poisson’s ratio was used (Figure 2b).

Auxetic particles have also been found to slightly deform in composites; however, the mechanism underlying that small deformation differs between the matrix and the auxetic material. A stiff particle composite has limited stretch due to its high modulus. When part of the matrix is replaced by a high modulus component, the stress is observed at a lower deformation than in a matrix. In an auxetic material, the resulting deformation is the result of two forces acting at the matrix–particle interface. The matrix tends to compress the particle, and the particle tends to expand in the opposite direction. These forces are likely to be disrupted. The same mechanism can be observed in Figure 1.

Different mechanisms could be observed in the path of load transmission. In the composite, the load was transmitted by the nearest line between the particles (Figure 2a). In the auxetic particles, however, the load was more directly and closely transmitted to the edge of the particles.

Figure 2a,b show a cross-section of a macroscopic model of a box with 172 particles; part a shows a solid particle composite, whereas part b shows an auxetic particle composite. The boxes were deformed to the z-stress of the same average; the average value is highlighted on the color map. The green rectangle shows the cross-section of a deformed matrix. The lines are the traces of the maximum transmission of the load inside the material.

The presence of heterogeneities in composites affects the response of the materials to external mechanical stress. The reinforcement mechanism is theoretically well-described in a composite material as dependent on its significantly stiffer particle than the matrix. The data in this paper (Table 1) were analyzed with the MT (Equations (1)–(10)) and self-consistent field mechanics (SCF) (Equations (11)–(18)) theories.

The main result was that the strain intensity tensor part of the intensity related to the particle.

For the rigid particle, E_m_ = 2.5kPa, E_p_ = 80kPa, and υ_m_ = υ_p_ = 0.39:(19)$$T^{rigid}=\left[\begin{array}{cccccc} 0.0584 & –0.0088 & –0.0088 & 0 & 0 & 0\\ –0.0088 & 0.0584 & –0.0088 & 0 & 0 & 0\\ –0.0088 & –0.0088 & 0.0584 & 0 & 0 & 0\\ 0 & 0 & 0 & 0.0672 & 0 & 0\\ 0 & 0 & 0 & 0 & 0.0672 & 0\\ 0 & 0 & 0 & 0 & 0 & 0.0672 \end{array}\right]$$

and for the auxetic particle, E_m_ = E_p_ = 2.5kPa, υ_m_ = 0.39, and υ_p_ = –0.9:(20)$$T^{auxetic}=\left[\begin{array}{cccccc} 1.1265 & 1.1249\; & 1.1249\; & 0 & 0 & 0\\ 1.1249\; & 1.1265 & 1.1249\; & 0 & 0 & 0\\ 1.1249\; & 1.1249\; & 1.1265 & 0 & 0 & 0\\ 0 & 0 & 0 & 0.0016 & 0 & 0\\ 0 & 0 & 0 & 0 & 0.0016 & 0\\ 0 & 0 & 0 & 0 & 0 & 0.0016 \end{array}\right]$$

The stiffness was detected by the first three diagonal elements that were positive: thus, the particle would increase the modulus both in rigid and auxetic particle composites, which was observed in the increased value of the effective modulus with a certain volume fraction of particles (Figure 3). The value of Poisson’s ratios are presented in Appendix A.

In this article, we selected the convergence criterion from Equation (18) (i.e., tol = 10^−6^). The solution of SCF converged after approximately 2500 iterations. The solutions of the MT and SCF methods were found to be nearly identical. The solution of the MT model corresponded to the solution of the SCF model after the first iteration. After the first few iterations, SCF significantly approached the final solution. The next iterations were small corrections with minimal intervals.

The effective stiffness tensor was calculated from the matrix using Equations (8)–(10) (MT model) or Equations (14)–(18) (self-consistent mechanics model). The effective stiffness tensor (**C**) was applied to calculate the material constants E or υ. 

The symmetric part of matrix C was calculated as:(21)$$C_{\mathrm{SYM}}=0.5(C+C_{\prime })$$
where **C′** is the transpose matrix of the stiffness tensor. The compliance matrix (S) was calculated as an inverse matrix of the normal part of **C_SYM, 1:3, 1:3_**_._ The value **C_SYM,1:3, 1:3_** refers to a symmetric matrix, which was related to normal deformation and was a submatrix of the first three columns and rows of the symmetric matrix: S = inv (**C_SYM, 1:3, 1:3_**)(22)

In the case of the isotropic material, the moduli were equivalent in all directions: E_x_ = E_y_ = E_z_ = 1/S_1,1_ = 1/S_2,2_ = 1/S_3,3_(23)

The shear moduli were calculated as:E_xy_ = E_xz_ = E_yz_ = 0.5 × C_4,4_ = 0.5 × C_5,5_ = 0.5 × C_6,6_(24)

Poisson’s ratio was calculated from the diagonal matrix:(25)$$E_{D}=\left(\begin{array}{ccc} E_{x} & 0 & 0\\ 0 & E_{y} & 0\\ 0 & 0 & E_{z} \end{array}\right)$$

The matrix of Poisson’s ratios was calculated from the diagonal matrix and compliance:**υ****_matrix_** = −E_D_…S(26)

In the case of the isotropic materials, all of the Poisson’s ratios were equivalent:υ = υ_matrix,1,2_ = υ_matrix,1,3_ = υ_matrix,2,1_ = υ_matrix,2,3_ = υ_matrix,3,1_ = υ_matrix,3,1_(27)

The Poisson’s ratios are presented in the Appendix A.

The first index is the main loading direction, and the second index is the perpendicular direction shrinking.

In this article, we provided a model for the isotropic material with a simple spherical inclusion and unity aspect ratio of particles. It is possible to model other materials, in which case the values in Equations (23), (24) and (27) are not equivalent.

In the case of the isotropic material with φ_p_ = 30% volume, the MT or SCF models’ solutions comprised two values: the modulus and Poisson’s ratio. The results from the solid particle composites were E = 4.8 kPa and υ = 0.36.

The data from the MT models could be applied to the projection of the expected tensile curve; the curve was calculated from the deformation of the homogenized box with the aforementioned elastic modulus and Poisson’s ratio calculated from the MT model (Figure 4a). The tensile curves for all volume fractions are presented in Appendix A. At the same time, the tensile curve was calculated for the heterogeneous FEM model from Figure 2a. As expected, both the MT and FEM models presumed that the material had a more reinforcing effect than the matrix. The curves were not aligned because the MT and SCF models constituted the stress on the boundary of the infinite box with the inclusion of specific volume fractions, whereas the FEM tensile curve was the average stress in the volume of the box.

The same calculations were performed for the materials with auxetic particles with φ_p_ = 30% volume (Figure 4b). The model from Figure 2 was macroscopically isotropic and, in the case of the solid particle composites, the results were E = 3.9 kPa and υ = 0.18. In this case, the presence of auxetic particles reinforced the material as a result of both the MT and FEM models, although the curves did not overlap for the same reason as in the composites.

The reinforcing effect in both the solid and auxetic particle composite materials was due to non-deformable particles; thus, both materials should be more reinforced than the matrix. The solid particle composites were more loaded than the auxetic particle material, most likely because the stress in the auxetic material was more evenly distributed than in the solid particle composites.

## 3. Discussion

The subject of this paper is based on materials that have auxetic spherical inclusions and conventional matrices. The materials can be discussed from two aspects.

### 3.1. Reinforcing Effect of Auxetic Spherical Inclusion

Reinforcement is the main aspect of such types of gels. The reinforcing effect was observed in both simulations, i.e., models based on micromechanics (Figure 3) and FEM simulations (Figure 4). The samples that are presented in this paper are model materials, which are intended to demonstrate the effect. Those materials will be never used in material design; however, one can imagine more sophisticated materials based on these principles, which will be suitable in the design of real materials.

The advantage is the design of a programmable structure, which can be tailored to the demands of the customer. The material can be designed in order to have the desired stiffness and shrinking in a lateral direction. The disadvantage is that it is still impossible to produce microscopic and nanometer-* sized auxetic particles. It is the main limitation of the construction of optical materials.

In comparison with the standard methods such as additional crosslinking filling by (nano)particles, auxetic material for the construction of reinforced materials is still not used in practical applications. This pilot study shows that there can be a potential effect, which can be suitable for some applications.

### 3.2. Distribution of Stress

Auxetic particle materials have a specific distribution of internal stress in comparison with standard rigid particle composites (Figure 1 and Figure 2). A standard composite can be simplified as a system of two springs: soft and stiff. It is a partly parallel (Maxwell) and partly serial (Kelvin Voigt) model. According to the simple models, the particles are mechanically loaded, and the stress is detected inside the particle and is mainly located in the center of the particle. By contrast, an auxetic phase controls the distribution of stress on the particle–matrix interface. The external force applied to the hybrid material with the highest local stress is on the interphase. This enables one to design a material that is highly resistant to an external force. However, inside the space shielded from external mechanical stress, the space is ready for a protective function; for example, some capsules of active compounds which must remain intact during manipulation.

## 4. Conclusions

The popularity of auxetic materials has grown among the research community despite the fact that their application remains sporadic. Combining auxetic and conventional materials into one mechanical unit may be an avenue for future applications. Recently, layered and sandwiched structures composed of an auxetic layer and one or more conventional layers have been presented in the literature, but more complex systems are missing. To date, studies have described the systems intuitively rather than analytically. There are no publications on stress analysis and its distribution on the interphase. This paper has provided an analytical description of an auxetic/conventional gel material from the point of view of stress analysis. The investigated material was a composite with auxetic heterogeneities incorporated into a conventional material as spherical particles. The deformation was analyzed in terms of micromechanics using MT and SCM models. The materials were structurally similar to composite gels filled with rigid particles, and they had a similar deformation response. However, in the case of auxetic particles, the stress was also more equally distributed in the space. Auxetic particles are promising new components for designing new materials and are capable of reinforcing a material without being affected by high densities.

## 5. Materials, Models, and Methods

### 5.1. Model-Finite Element Method

The deformation behavior was modeled by a finite element analysis, which was designed using the GIBBON [56] (Kevin M. Moerman, Massachusetts Institute of Technology, Cambridge, MA, USA) software package; the deformation was applied by the FEBIO software suite [57,58] (version 2.9.1, Musculoskeletal Biomechanics Laboratory, Columbia University, New York, NY, USA). 

### 5.2. Detailed Model with Real Auxetic Configuration

The first step was to design a representative material box in which the honeycomb structure was formed from a representative cube block in the xz projection (see Appendix A). The same honeycomb pattern was also formed from the vertical direction (yz plane). The result was a 3D structure with a honeycomb projection in two directions. In the first step, the structure was composed only of conventional honeycombs.

Part of the cells inside the spherical inclusion was transformed into a reentrant honeycomb. A combined structure of an auxetic inclusion and conventional matrix was designed (Figure 1a). The honeycomb structure was demonstrated in a slice of the box (Figure 1b). The model of the volume was discretized to the cubic mesh.

The model’s upper boundary had a prescribed displacement of up to 20% relative elongation in 20 steps. The lower boundary of the sample was fixed, whereas the other boundaries were free.

### 5.3. Model of Multiple Particles in a Cubic Box

A 172-sphere model was proposed. The centers of the spheres were arranged in a face-centered cubic (FCC) mesh, whose basic unit cells were 3 × 3 × 3. The inclusions were distributed in a cubic box so that the spheres in the box had a certain volume fraction. The result was a parameterized cubic box with solid spherical particles.

The parameterization of the model was coarser. The inclusions and matrix had prescribed deformation parameters: elastic modulus, Poisson’s ratio, and density. The matrix was a conventional material with a positive Poisson’s ratio and a particle with a negative value. There was absolute adhesion on the interphase between the particles and the matrix. The model was discretized by a 3D triangular network. The lower boundary was fixed, and the upper boundary had a prescribed displacement of up to 30% relative elongation of the box in 20 steps.

### 5.4. All Simulated Materials Were Modeled as Neo-Hookean Solid Bodies

The adjustable parameters of the neo-Hookean model material were (1) material density, (2) elastic modulus, and (3) Poisson’s ratio. The density of all materials was set as 1 gcm^−1^. The moduli and Poisson’s ratios varied depending on the type of model (Table 1).

The single-particle model material for the 3D honeycomb structure was monolithic in the entire sample. There was no projected material interphase. The boundary between the inclusion and the matrix was only in the structure of the conventional/reentrant honeycombs;The heterogeneous material with the 3D honeycomb had a soft matrix (Item i.). The model was heterogeneous, and the modulus of inclusion was 40-fold stiffer than the matrix modulus. The high modulus led to rigid inclusion;A neat matrix in a coarse model was modeled from the material with Poisson’s ratio of zero because there was no stress concentration in any material. The zero value of Poisson’s ratio was more or less similar to the weighted average value of a 30% composite gel with auxetic inclusion;The composite material that included auxetic particles had a homogenous modulus in the entire sample: matrix and inclusion. However, the matrix had a positive Poisson’s ratio, and the inclusions had a prescribed negative Poisson’s ratio;In the standard solid particle composite gel, the particle had a 40-fold higher modulus than the matrix. The particles were nearly undeformable. In case 2, the Poisson’s ratio of the particle played no role.

FEBIO was the software used to record all data, including the stress of the elements and displacement of the nodes. The results recorded in the FEBIO output enabled the reconstruction of the deformation simulation.

We recorded the z-stress (i.e., the stress in the z-direction acting on the upper plane with a z-normal vector) using FEBIO software. The values of the Cauchy z-stress across the upper boundary of the sample were the tensile curve of the virtual sample. The stresses in other directions were significantly smaller than the z-stress.

For the step-by-step constructions of both models, please see Appendix A. For more details please contact main author. The scan of the modeling box is presented in animation (Appendix A).

## Figures and Tables

**Figure 1 gels-08-00698-f001:**
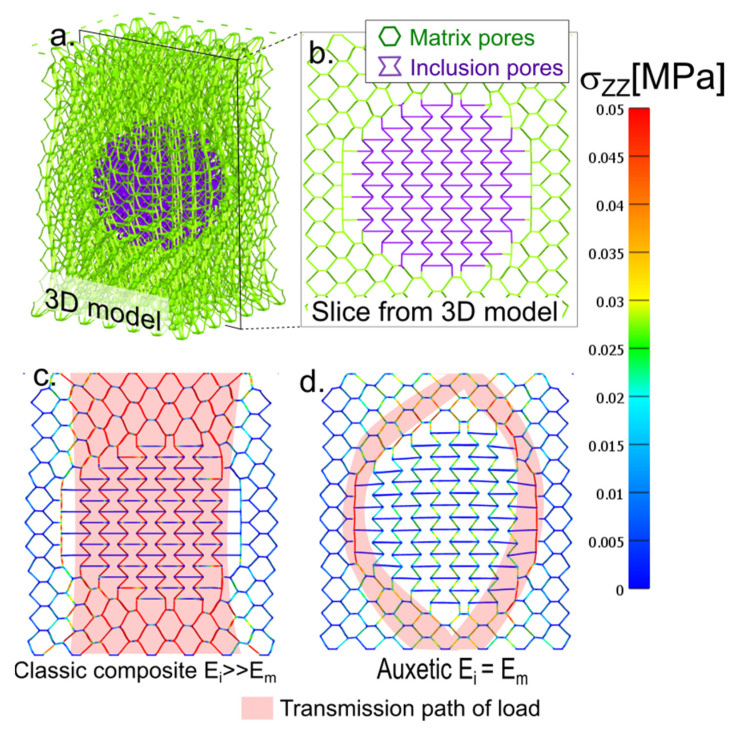
Combined auxetic and conventional hexagonal honeycomb structure: (**a**) 3D model of one particle; (**b**) one slice of the 3D model, violet represents the auxetic inclusion, and green represents the conventional matrix; (**c**) model of the solid particle composite gel deformed in the z-axis. The trace of load transfer inside the material is highlighted in light red; (**d**) auxetic inclusion gel in the conventional matrix colored axis and z-stress (z component of stress acting on the plane with a z-normal vector).

**Figure 2 gels-08-00698-f002:**
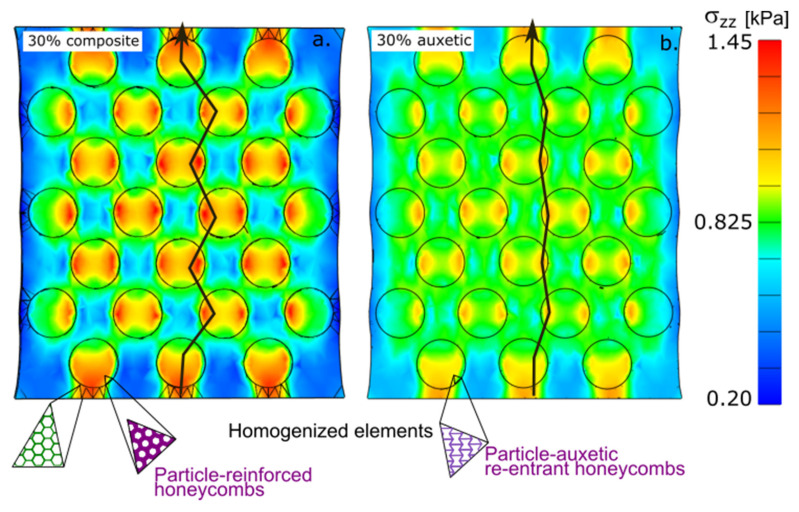
Cross-section of a macroscopic model of a box with 172 particles: (**a**) rigid particle composite gel and (**b**) auxetic particle gel.

**Figure 3 gels-08-00698-f003:**
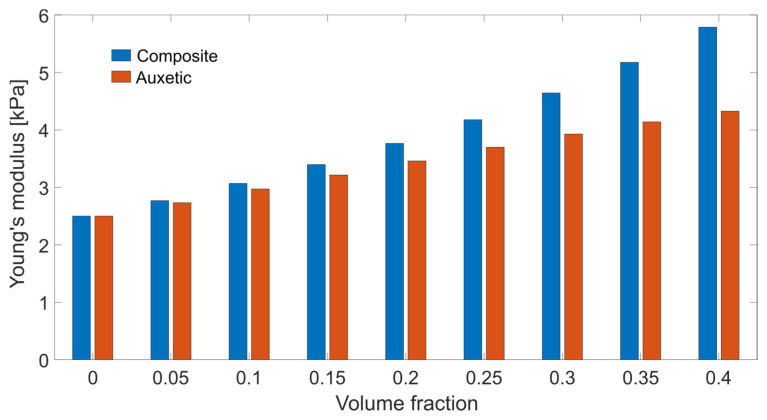
Effective modulus from the Mori–Tanaka model as a function of the volume fraction of the particles.

**Figure 4 gels-08-00698-f004:**
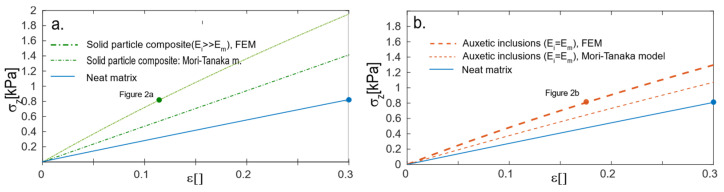
The tensile curve of the matrix, Mori–Tanaka model (thin), and FEM model of the heterogeneous material (thick). The macroscopic box consists of 172 particles in a cubic box distributed to face-centered cubic: (**a**) solid particle composite and (**b**) auxetic particle composite. Points: stretching ratio of the model cross-section in reference to Figure 2.

**Table 1 gels-08-00698-t001:** List of models, E_m_ and E_i_ modulus of the matrix and inclusions, and υ Poisson’s ratio of the matrix and inclusions. ^a^ The auxetic effect was achieved by the structural configuration of the material. * Matrix is porous-modulus and Poisson’s ratio are parameters of solid phase ^△^ Particle is porous: modulus and Poisson’s ratio are parameters of solid phase.

Model Name	E_m_ [kPa]	E_i_ [kPa]	υ_m_	υ_i_	Description
3D-cellular auxetic ^a^ inclusion	2.5 *	2.5 ^△^	0.49 *	0.49 ^△^	3D-cellular conventional */reentrant ^△^ honeycomb structure
3D-cellular undeformable inclusion	2.5 *	100 ^△^	0.49 *	0.49 ^△^	-
Matrix	2.5	-	0.00	-	Neat matrix with zero transversal deformation
Auxetic inclusion	2.5	2.5	0.39	–0.9	Material with homogenous rigidity inclusion is auxetic
Composite inclusion	2.5	100	0.39	0.2	Standard composite: soft matrix, rigid particle

## Data Availability

Data can be provided by main/corresponding author including g-codes for the printing of experimental samples.

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
