# Peer review of "Deformation of Gels with Spherical Auxetic Inclusions"

_gels, 2022, doi:10.3390/gels8110698_

Round 1

Reviewer 1 Report

This manuscript reports the deformation of the gels with inclusions. The topic is reasonable. I recommend publishing after the following issues are solved.

1, To solidify the background of the gels in deformation, I recommend citing more papers on this point, such as Journal of Applied Physics, 123, 034505 (2018).

2,  How to identify the draft of figure 1? Any reference to support this?

3, Any experimental demonstration for Figure 2?

4, How to calculate Young's modulus of Figure 3?

Author Response

This manuscript reports the deformation of the gels with inclusions. The topic is reasonable. I recommend publishing after the following issues are solved.

We thank the reviewer for the constructive criticism. We tried to include his/her valuable comments to the paper.

1, To solidify the background of the gels in deformation, I recommend citing more papers on this point, such as Journal of Applied Physics, 123, 034505 (2018).

We thank very much for this comment. Indeed, strain sensing can be also the new application of hybrid-auxetic. The references were added to the page 2 paragraph 3

2, How to identify the draft of figure 1? Any reference to support this?

Sorry, I did not exactly understand what is required. The Figure 1 is a result of our simulations. It is referred in the text page …. paragraph….  Please specify the question. We found that there is not the annotation and scale of colorbar in Figure 1 - we added it to the Figure 1.  We tried to improve Figure 1 in order to increase the clarity.

3, Any experimental demonstration for Figure 2?

We did only the simple experimental tensile test. We plan to do the optical (video) extensiometry. We plan to do it in the near future externally. We do not have the video extensimeter in our department.

4, How to calculate Young's modulus of Figure 3?

We added the calculation of the materials constants from the compliance matrix (page 9, Paragraph 8)

Reviewer 2 Report

This manuscript should be re-written carefully to increase the clarity of its statements. Just as examples:

 Abstract: “offering to introduce interesting features” should read “offering interesting features”.

Abstract: “multi-particles face-centered cubic (FCC) lattice”. Do the authors mean “multi-particles face-centered WITH cubic (FCC) lattice”?

Absreact, Line 14: “provide” should read “provides”.

Abstract is very general without giving the critical information and data. For example, it says “inclusions embedded in a conventional matrix”. Matrix should exactly be specified. Also general statements should be supported by specific values to provide meaningful information.

‘Introduction, line 31: “This paper aims to provide a detailed description of the micromechanics of these composites”. This is yet another unclear statement. Micromechanics of what?!

Page 3, line 107: ”Several examples can be found of composites with unique structural properties.” This statement does not provide any meaning!

Page 3, line 110: “The overview of methods is proposed to increase orientation in article”. Do the authors mean “An overview of methods employed in this article is provided to further indicate the orientation of the research”?

At the current situation, I do not recommend this article for publication.

Author Response

This manuscript should be re-written carefully to increase the clarity of its statements.

We are very thankful for careful reading and deep analysis of the text of manuscript. 

I am sorry, I (the first author) am not English native speaker and my English is moderate. I asked the proofreader and discussed the modifications. The proofreader (American English native speaker with similar professional background) did the complex proofreading: grammar, style and overall comprehensibility of document. According to him, the text is now comprehensible. It should be more or less without grammatical errors. 

The results of proofreading are highlighted in revised file. I can send the certificate of proofreading.

In fact the main statement we want to communicate that there is some combination of conventional matrix and auxetic particle can be a new promising method for reinforced material. In this paper, we describe the model material where the auxetic and conventional phase are from the same material and their properties were tuned by porosity.

On the other hand there can be designed much more interesting material with matrix from solid (non-porous) gel and particle from some durable material. That material can be lightweight and, at the same time, mechanically resistant and can have shape stability. Such material (as we think) was still not in literature. And it will be relatively simply tuned because the stiffness of gel in a matrix can be finely tuned by slight additional covalent crosslinking. The stiffness of auxetic material can be tuned by degree of porosity of reentrant cells. The materials was still not produced but we plan to do it, because in the next year we will have 3D print from gel.

The next interesting property is distribution of deviatoric stress in our materials, which we observed in our models. It is not included in the paper, but one can easily discover it, when one will reproduce the model. (We have data but finally we did not include because it is a passage from continuum mechanics difficult to explain without experiment.). In conventional composites the maximum deviatoric stress is directly on the surface of particle. It has a consequence the dissociation of particle from matrix, because there is shearing of matrix on the particle surface. It is weak point of the most common composites. In our hybrid auxetic materials, the deviatoric stress is directed to the layer under surface of particle material. If the particle material will be extremely durable, the composite material will become more resistant to the failure.  

Just as examples: Comments:

 Abstract: “offering to introduce interesting features” should read “offering interesting features”.

It was modified.

Abstract: “multi-particles face-centered cubic (FCC) lattice”. Do the authors mean “multi-particles face-centered WITH cubic (FCC) lattice”?

Modified

Abstract, Line 14: “provide” should read “provides”.

Modified

Abstract is very general without giving the critical information and data. For example, it says “inclusions embedded in a conventional matrix”. Matrix should exactly be specified. Also general statements should be supported by specific values to provide meaningful information.

Abstract was extended

‘Introduction, line 31: “This paper aims to provide a detailed description of the micromechanics of these composites”. This is yet another unclear statement. Micromechanics of what?!

The part of micromechanics was completed.

Page 3, line 107: ”Several examples can be found of composites with unique structural properties.” This statement does not provide any meaning!

The sentence was deleted

Round 2

Reviewer 1 Report

The author solved all my issues. I recommend publishing in the current form.

Reviewer 2 Report

The revised article may be considered for publication.